# Incorrect Feeding Practices, Dietary Diversity Determinants and Nutritional Status in Children Aged 6–23 Months: An Observational Study in Rural Angola

**DOI:** 10.3390/children10121878

**Published:** 2023-11-30

**Authors:** Andrea Pietravalle, Alessia Dosi, Telmo Ambrosio Inocêncio, Francesco Cavallin, Joaquim Tomás, Giovanni Putoto, Nicola Laforgia

**Affiliations:** 1Doctors with Africa CUAMM, 35121 Padua, Italy; 2Doctors with Africa CUAMM, Luanda 56918-000, Angola; 3Missionary Catholic Hospital of Chiulo, Ombadja 23030, Angola; 4Independent Statistician, 36020 Solagna, Italy; 5Section of Neonatology and Neonatal Intensive Care Unit, Interdisciplinary Department of Medicine (DIM), University of Bari “Aldo Moro”, 70121 Bari, Italy

**Keywords:** malnutrition, dietary diversity, incorrect feeding practices

## Abstract

Background: More than a quarter of children who are affected by severe acute undernutrition reside in Sub-Saharan Africa. Incorrect feeding practices have a negative impact on a child’s health in both the short and the long term, and the interval from conception to two years is the most critical for the development of undernutrition-related complications. These first 1000 days of life also represent an “opportunity window” for early interventions, hence, having a clear insight into dietary habits and the determinants of diet quality is fundamental to improving nutritional counseling practices. Objectives: To investigate incorrect feeding practices, dietary diversity determinants and nutritional status in children aged 6–23 months. Methods: Prospective quali-quantitative observational study conducted at the Missionary Catholic Hospital of Chiulo (Angola) from March to April 2023. Results: Of 250 children, global acute malnutrition affected 25.2% and was associated with starting complementary feeding at <4 months of age (*p* = 0.007) and not achieving the minimum meal frequency (*p* < 0.0001). Minimum dietary diversity was found in 11.2%, minimum meal frequency was experienced by 72.8%, and the minimum acceptable diet was found in 11.2% of participants. The minimum dietary diversity was reached only by households with access to food from five or more major food groups (*p* = 0.007) or the money to buy food from five or more major food groups (*p* = 0.008) and was higher in households where the householder had a higher educational level (*p* = 0.002). Regarding the determinants linked to family traditions and beliefs, the main religion-associated beliefs concerned the impurity of pork (*n* = 25) and fish (*n* = 8), while eggs (*n* = 19) and cow milk (*n* = 8) were the main food types that were deemed harmful for children. Conclusions: Although some factors (economic and religious) may be difficult to overcome, other factors linked to erroneous beliefs (dangerous foods) or incorrect feeding practices (early weaning and an incorrect frequency of meals) can be targeted, to improve the effectiveness of nutritional counseling practices.

## 1. Introduction

The word “malnutrition” may denote undernutrition or overnutrition status, but it usually means undernutrition, which includes acute (wasting), chronic (stunting) and composite forms, according to the degree and timing of nutritional deficiency [1]. 

According to recent estimates, 13.6 million children under the age of five globally are affected by severe acute undernutrition, and more than a quarter of these live in Sub-Saharan Africa [1].

The early interruption of breastfeeding and the early start of weaning, together with the assumption of a quantitatively and qualitatively poor diet, have a negative impact on a child’s health both in both the short (mortality) and the long term (morbidity, mental skill, educational success, work proficiency and income) [2].

The first 1000 days of life (from conception to 2 years) represent the most critical period for the development of undernutrition-related complications, but also represent an “opportunity window” for early intervention [3]. Dietary counseling is a key aspect in the management of malnourished children and should be included in the primary aspects of the treatment strategy but is often absent or inadequate [4]. Understanding dietary habits and the determinants of diet quality is fundamental to improving nutritional counseling practices. Dietary diversity, based on the evaluation of different food groups consumed daily, is a key indicator of diet quality and nutrient adequacy in infants aged 6–23 months [5]. An insight into the burden and role of the determinants of inadequate dietary diversity represents a crucial starting point for the improvement process [6].

This study aimed to investigate the incorrect feeding practices, nutritional status and dietary diversity determinants of children aged 6–23 months, who were admitted to the Missionary Catholic Hospital of Chiulo in Angola.

## 2. Materials and Methods

This is a prospective quali-quantitative observational study conducted at the Missionary Catholic Hospital of Chiulo from March to April 2023. The Ethics Committee of the Angolan Ministry of Health approved the study (ref. number 6/C.E.M.S./2023). Informed consent was obtained from all subjects involved in the study. All procedures were undertaken according to appropriate guidelines and regulations.

### 2.1. Setting

The Missionary Catholic Hospital of Chiulo (Cunene province, Angola) is a district hospital implementing the Community Management of Acute Malnutrition program in a rural area of 12,263 km^2^ with 345,490 inhabitants (including 60,392 children under 5 years) [4].

Within a network of 36 healthcare facilities implementing the national nutrition program, the hospital acts as a stabilization center for the inpatient care of malnourished children with complications and as outpatient treatment units for the rehabilitation phase after discharge.

### 2.2. Definitions

*Malnutrition* is defined by the combination of clinical evaluation and anthropometrical measurements (Weight for Height ratio or Mid-Upper Arm Circumference) according to the classification of the World Health Organization [7]. *Severe Acute Malnutrition* (SAM) is indicated by Weight for Height ratio < 3 Standard Deviation and Mid-Upper Arm Circumference ≤115 mm. *Moderate Acute Malnutrition* (MAM) is indicated by Weight for Height ratio ≥ 3 and <2 Standard Deviations, or a mid-upper arm circumference between >115 and <124 m.

*Global acute malnutrition (GAM)* is the proportion of children aged 6–59 months, in a given population, with either SAM or MAM [8].

*Minimum dietary diversity (MDD)* is the proportion of children who ate food and beverages from at least five of these eight different food groups in the day before the evaluation: 1. breast milk; 2. grains, roots, tubers and plantains; 3. pulses, nuts and seeds; 4. dairy products; 5. flesh foods; 6. eggs; 7. vitamin-A rich fruits and vegetables; and 8. other fruits and vegetables [9].

*Minimum meal frequency (MMF)* is the proportion of children who ate in the day before the evaluation at least: 1. two feedings of solid, semi-solid or soft foods for breastfed infants aged 6–8 months; 2. three feedings of solid, semi-solid or soft foods for breastfed children aged 9–23 months; 3. four feedings of solid, semi-solid or soft foods or milk feeds for non-breastfed children aged 6–23 months with at least one of the four feeds as a solid, semi-solid or soft feed [9].

*Minimum Acceptable Diet (MAD)* is the percentage of children who consumed, on the previous day of evaluation, a minimum acceptable diet (achieving both MDD and MMF) [9].

### 2.3. Patients

All children aged 6–23 months admitted within the study period, were eligible for inclusion.

### 2.4. Outcome Measures

The outcome measures of the quantitative analysis included GAM, MDD, MMF and MAD. Family traditions and beliefs were included in the qualitative analysis.

### 2.5. Data Collection

Data were collected and recorded anonymously by using a form that was submitted to caregivers and consulting hospital charts. Data collection included: (a) children’s data: age, sex, birth weight, height/length and nutritional status; (b) known risk factors for malnutrition: maternal age, educational status, number of pregnancies, birth weight, duration of exclusive breastfeeding and age of complementary feeding introduction; (c) dietary habits: MDD, MMF and MAD; and (d) known dietary diversity determinants: income, cash availability, socio-economic status, agriculture and agrobiodiversity, gender, household size and family traditions.

### 2.6. Statistical Analysis

Descriptive analyses were reported as median and interquartile range (IQR) (numerical variables) or absolute frequency and percentage (categorical variables). Associations between categorical variables were assessed using the Chi square test or Fisher’s test. All tests were 2-sided and a *p* of <0.05 was considered statistically significant. Statistical analysis was carried out using R 4.3 (R Foundation for Statistical Computing, Vienna, Austria) [10].

## 3. Results

The study included 250 children (132 male and 118 female) aged 6–23 months (Table 1). There were 43 with MAM (17.2%) and 20 with SAM (8.0%), yielding a GAM of 25.2%.

The known risk factors for malnutrition are summarized in Table 2. GAM was present in 20.6% (18/87) of children who received exclusive breastfeeding at >6 months and in 27.6% (45/163) of those who received exclusive breastfeeding at <6 months (*p* = 0.29). GAM was present in 50.0% (12/24) of children who started complementary feeding at <4 months and in 22.6% (51/226) of those who started complementary feeding at >4 months (*p* = 0.007).

Overall, MDD was present in 11.2% of children (28/250), MMF in 72.8% (182/250) and MAD in 11.2% (28/250). The data on MDD and MAD overlapped. GAM was present in 10.7% (3/28) of children who achieved minimum dietary diversity and in 27.0% (60/222) of those who did not (*p* = 0.10). GAM was present in 12.1% (22/182) of children who achieved minimum meal frequency and in 60.2% (41/68) of those who did not (*p* < 0.0001).

The known determinants of MDD are described in Table 3. MDD was reached only by households with access to food from five or more major food groups (*p* = 0.007) or income to buy food from five or more major food groups (*p* = 0.008). Notably, the data suggested that 197/250 households (78.8%) had access to food from five or more major food groups, but only 28/197 (14.2%) fed them to children; similarly, 198/250 households (79.2%) had the income to buy food from five or more major food groups, but only 28/198 (14.1%) fed them to children. In addition, MDD was higher in households where the householder had a higher educational level (19.3% vs. 5.9%, *p* = 0.002).

Family traditions and religious beliefs cause the elimination of some foods:-Eggs, because of dumbness (*n* = 18) or diarrhea (*n* = 1);-Cow milk, because of wounds, according to the Kimbanguist religion (*n* = 8)-Fish, because it hurt the skin, according to the Kimbanguist religion (*n* = 2), or is harmful for children (*n* = 1);-Meat (*n* = 1) because it is harmful for children;-Vegetables (*n* = 1) because they are harmful for children;-Pork, because it is impure (*n* = 25) (Isaia 66, 17; Levitico 11) (Kimbanguist religion);-Fish without scales, because it is impure (*n* = 8) (Kimbanguist religion);-Fish, because it is holy (*n* = 1);-Meat, because it is holy (*n* = 1).

## 4. Discussion

Attaining ideal nourishment during the first 2 years of life is critical because it reduces morbidity and mortality, lowers the risk of chronic disease, and fosters better development overall [10]. The WHO and UNICEF recommendations include: beginning breastfeeding within 1 h of birth; exclusive breastfeeding for the first 6 months of life; and the introduction of nutritionally adequate and safe complementary (solid) foods at 6 months, together with continued breastfeeding up to 2 years of age or beyond [11,12].

Breastmilk is a significant source of energy and nutrients in children aged 6–23 months, and exclusive breastfeeding up to 6 months offers several advantages for the infant, with a significant protection against gastrointestinal infections [11]. On the other hand, if complementary foods are not introduced in a timely manner, when the need for energy and nutrients exceeds what is provided by breast milk, growth can be impaired. However, the results of giving semi-solid foods to a still-immature child—jointly with poor food quality—are equally harmful [12]. Our data suggested an association between GAM and starting complementary feeding before 4 months of life, but no association was found with exclusive breastfeeding for less than 6 months. Of course, the role of other potential confounding factors, and the complex relationship between exclusive breastfeeding and introducing complementary feeding, suggest caution when drawing implications from such findings.

We found that only around one out of ten children achieved the minimum acceptable diet, in agreement with figures from the latest National Demographic and Health survey, which reported a 13% incidence of MAD [13]. Our data show that MMF was significantly associated with lower GAM, while MDD was not. These findings were in broad agreement with a recent study from the same setting, suggesting an association between lower GAM and a higher achievement of both MMF and MDD [14].

In our study, we investigated the prevalence and the role of the different determinants of dietary diversity, on the basis of the detailed classification described in a recent qualitative ethnographic study [15]. As previously suggested [6], we found that children living in a household with higher level of education had more chance of achieving minimum diet diversity, hence highlighting the importance of parental education. As expected, we found that minimum diet diversity could only be achieved when there was access to food from five or more major food groups and enough income to buy these foods. Noteworthily, most households had access to food from five or more major food groups and adequate income, but only a minority of them fed the children with food from five or more major food groups. The possibility that cultural and/or religious beliefs play a role seems not to be the reason, because our data demonstrate that MDD was similar in households with and without family traditions and beliefs. Nonetheless, the qualitative analysis highlighted some religion-associated beliefs on food impurity (mainly regarding pork and fish) and other beliefs about specific foods being harmful for children (mainly regarding eggs and cow milk). These data are in line with findings from similar settings and should be considered when planning educational interventions to improve child nutrition [16].

A recent work underlined sociocultural impacts on food preferences and the role of analyzing cultural food routines when modelling appropriate and successful policy interventions [17]. Overall, we acknowledge that economic and religious constraints may be difficult to overcome, but other factors such as erroneous beliefs (i.e., about dangerous foods) and incorrect feeding practices (early weaning and incorrect meal frequency and composition) may be useful targets to reach with appropriate educational interventions. Specific education on these elements should find a place within the social and behavioral programs, known as “parenting interventions”, that focus on enhancing caregivers’ knowledge, attitudes, practices and skills, with the final goal of endorsing the best early child development [18].

Our study has some limitations. First, the study design limits the generalizability of the results to different settings. Second, the sample size is limited. Third, the available data did not allow for further investigations of some interesting findings, such as the reasons for not properly feeding children despite the availability of food and money, together with the roles of family traditions and beliefs.

## 5. Conclusions

The present study highlighted specific aspects that should be targeted by interventions aiming at improving the effectiveness of nutritional counseling practices, which represent a fundamental tool for prevention of severe childhood undernutrition.

## Figures and Tables

**Table 1 children-10-01878-t001:** Child characteristics.

*n*	250
Age (months) (median (IQR))	12 (9–15)
Male/Female	132:118
Weight (kg) (median (IQR))	8.3 (7.0–9.2)
Height/length (cm) (median (IQR))	71 (67–75)
Edema (*n* and %)	
absent	225 (90.0)
+	15 (6.0)
++	10 (84.0)
Nutritional status (*n* and %)	
Normal	187 (74.8)
MAM	43 (17.2)
SAM	20 (8.0)

**Table 2 children-10-01878-t002:** Known risk factors for malnutrition.

Mothers	
Age (years) (median (IQR))	27 (22–32)
Number of pregnancies (median (IQR))	3 (2–5)
Educational status (*n* and %)	
Never attended school	74 (29.6)
Primary school	79 (31.6)
Secondary school	97 (38.8)
Children	
Birth weight (kg) (median (IQR)) *	3.2 (2.9–4.8)
Mother as primary caregiver (*n* and %)	244 (97.6)
Breastfeeding (*n* and %)	208 (83.2)
Duration of exclusive breastfeeding (*n* and %)	
<6 months	163 (65.2)
>6 months	87 (34.8)
Age of complementary feeding introduction (*n* and %)	
<4 months	24 (9.6)
4–6 months	139 (55.6)
>6 months	87 (34.8)

* Birth weight was available for 120/250 children.

**Table 3 children-10-01878-t003:** Known determinants of minimum dietary diversity (MDD).

Determinant	*n* (%)	MDD: *n* (%)	*p* Value
*Head of household*			
Gender:			0.99
Female	50 (20.0)	6 (12.0)	
Male	200 (80.0)	22 (11.0)	
*Educational status:*			0.002
Never attended/primary school	152 (60.8)	9 (5.9)	
Secondary school	98 (39.2)	19 (19.3)	
*Household size*			
Family members:			0.54
≤5	107 (42.8)	10 (9.3)	
>5	143 (57.2)	18 (12.6)	
Children < 5 years:		-	-
1–3	249 (99.6)		
>3	1 (0.4)		
*Agriculture, agrobiodiversity and livelihood diversity*			
Irrigated field:			0.16
No	234 (93.6)	24 (10.2)	
Yes	16 (6.4)	4 (25.0)	
Farm animals:			0.85
No	81 (32.4)	10 (12.3)	
Yes	169 (67.6)	18 (10.6)	
Householder with another job:			0.06
No	89 (35.6)	5/89 (5.6)	
Yes	161 (64.4)	23/161 (14.2)	
*Availability of foods*			
From ≥5 major food groups:			0.007
No	53 (21.2)	0 (0)	
Yes	197 (78.8)	28 (14.2)	
*Income*			
Ablility to buy food from =/> major food groups:			0.008
No	52 (20.8)	0/52 (0.0)	
Yes	198 (79.2)	28/198 (14.1)	
*Cultural aspect*			
Family traditions and beliefs			0.69
No	203 (81.2)	24/203 (11.8)	
Yes	47 (18.8)	4/47 (8.5)	

## Data Availability

The datasets used and/or analyzed during the current study are available from the corresponding author on reasonable request.

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
