# Peer review of "Incorrect Feeding Practices, Dietary Diversity Determinants and Nutritional Status in Children Aged 6–23 Months: An Observational Study in Rural Angola"

_children, 2023, doi:10.3390/children10121878_

Round 1

Reviewer 1 Report

Comments and Suggestions for Authors

In this prospective quali-quantitative observational study, the authors aimed to evaluate the impact of incorrect feeding practices on the nutritional status of children (6-23 months) and to investigate the role of dietary diversity determinants on their diet quality.
I recommend you move the definitions from Methods to Introduction, given that the Introduction is quite short.
Please also specify the exclusion criteria from the study.
The article presents 18 references, being up to date. I recommend that you add more references and thus expand the discussions, which are also rather short.
Otherwise, the article is well organized and clear.

Reviewer 2 Report

Comments and Suggestions for Authors

Dear authors,

Low diet quality and quantity and inadequate feeding practices can cause malnutrition. Poor nutritional status in early childhood is associated with stunted growth. From this point of view, the article has a scientific value, malnutrition continues to be a public health problem, especially in rural areas of poorly developed or developing countries.

Back to the article:

- the introduction must be extended by adding new references to the topic

- In the material and method section, it might have been useful to define malnutrition and, according to the biological criterion, the decrease of serum proteins, especially of albumin, considering that the association of edemas (kwashirkor, or marasmic kwashiorkor) was specified.

- Discussion - must compare the results obtained from the study with other results from the recent literature. The expansion of these sections will certainly add references to new articles.

Good luck!

Reviewer 3 Report

Comments and Suggestions for Authors

Malnutrition is a significant public health concern for children, particularly in Africa. This study aimed to assess the impact of incorrect feeding practices on the nutritional status of children aged 6-23 months and to investigate the role of various dietary diversity determinants on their diet quality. The study explored several critical factors, including breastfeeding and weaning age, while also delving into new aspects such as economic, religious, and belief-related factors.

Nevertheless, I have a few concerns that I believe the authors should address:

1)The study's title, ‘Impact of incorrect feeding practices on the nutritional status of children (6-23 months) and the role of dietary diversity determinants on their diet quality: an observational study in rural Angola,’  used language like 'impact' and 'determinant,' suggesting causal relationships. However, the authors did not intend to establish causal associations in this study. It's crucial to use caution in the language to avoid misleading readers.

2)In the Methods section, the primary outcome is 'malnutrition,' defined through a combination of clinical assessment and anthropometric measurements. The authors employed GAM (either SAM or MAM), but only used Chi Square or Fisher’s tests in the analyses. It would be beneficial to explain why more advanced statistical methods, like logistic regression, were not employed to investigate the associations between risk factors and malnutrition. Factors such as breastfeeding and weaning age are influenced by family socioeconomic status, mother’s age, BMI, and birth weight, among others. The current results may be affected by confounding variables, making it challenging to discern the genuine associations.

3)In Results, Table 2 provides percentages of risk factors. However, it's unclear where the data presented in the manuscript (Line 129-133, Line 137-141) regarding risk factors and GAM and MDD came from. It would be helpful to clarify and potentially re-conduct the analyses.

4)On Line 142, Table 3 discusses 'the determinants of MDD.' As mentioned earlier, it's important to refrain from using causal language when conducting exploratory analyses.

5)Birth weight is a crucial factor in understanding malnutrition. The authors should clarify how they intend to address this factor. Is it considered solely a risk factor, or could it play a role as a mediator in the relationships under investigation?

6)Due to methodological and statistical limitations, the discussion may not be adequately supported. For instance, the statement, 'Our data suggested an association between GAM and starting complementary feeding before 4 months of life, but no association was found with exclusive breastfeeding for less than 6 months,' may not be accurate, as the analyses did not consider other potential confounding variables, not to mention the complex relationship between exclusive breastfeeding and introducing complementary feeding.

Comments on the Quality of English Language

Please refrain from excessive use of abbreviations, and be sure to provide definitions upon their initial occurrence.

Round 2

Reviewer 2 Report

Comments and Suggestions for Authors

Dear authors,

thank you for the revised version of your article. It is improve, but I still think that the references can be extended. 

I don't have any others questions.

Author Response

We thank the reviewer for his suggestion, we carried out further literature review, unfortunately without finding further works useful to expand our discussion